**Data Availability Statement:** All relevant data are within the manuscript and its Supporting Information files

**Funding:** The study was supported by the German Center for Lung Research (Deutsches Zentrum für

# High prevalence of falsely declaring nicotine abstinence in lung transplant candidates

**Tobias Veit**[1☺], **Dieter Munker**[1☺], **Gabriela Leuschner**[1], **Carlo Mümmler**[1], **Alma Sisic**[2], **Teresa Kauke**[3], **Christian Schneider**[3], **Michael Irlbeck**[4], **Sebastian Michel**[5], **Daniela Eser-Valerie**[6], **Maximilian Huber**[6], **Jürgen Barton**[1], **Katrin Milger**[1], **Bruno Meiser**[2], **Jürgen Behr**[1], **Nikolaus Kneidinger**[1]*

**1** Department of Internal Medicine V, Comprehensive Pneumology Center(CPC-M), Member of the German Center for Lung Research (DZL), University of Munich, Munich, Germany, **2** Transplant Center, University of Munich, Munich, Germany, **3** Department of Thoracic Surgery, University of Munich, Munich, Germany, **4** Department of Anaesthesiology, University of Munich, LMU, Munich, Germany, **5** Clinic of Cardiac Surgery, University of Munich, LMU, Munich, Germany, **6** Department of Psychiatry and Psychotherapy, University of Munich, LMU, Munich, Germany

☺ These authors contributed equally to this work.
* nikolaus.kneidinger@med.uni-muenchen.de

## Abstract

Tobacco use after lung transplantation is associated with adverse outcome. Therefore, active smoking is regarded as a contraindication for lung transplantation and should be excluded prior to placement on the waiting list. The aim of the study was to compare self-reporting with a systematic cotinine based screening approach to identify patients with active nicotine abuse. Nicotine use was systematically assessed by interviews and cotinine test in all lung transplant candidates at every visit in our center. Patients were classified according to the stage prior to transplantation and cotinine test results were compared to self-reports and retrospectively analyzed until June 2019. Of 620 lung transplant candidates, 92 patients (14.8%) had at least one positive cotinine test. COPD as underlying disease (OR 2.102, CI 1.110–3.981; p = 0.023), number of pack years (OR 1.014, CI 1.000–1.028; p = 0.047) and a time of cessation less than one year (OR 2.413, CI 1.410–4.128; p = 0.001) were associated with a positive cotinine test in multivariable regression analysis. The majority of non-COPD patients (n = 13, 72.2%) with a positive test had a cessation time of less than one year. 78 patients (84.7%) falsely declared not consuming any nicotine-based products prior to the test. Finally, all never smokers were test negative. In conclusion, our data demonstrate that active nicotine use is prevalent in transplant candidates with a high prevalence of falsely declaring nicotine abstinence. COPD was the main diagnosis in affected patients. Short cessation time and a high number of pack years are risk factors for continued nicotine abuse.

## Introduction

Smoking is the main cause for chronic lung disease and is associated with high individual and socioeconomic burden. For selected patients, lung transplantation can be offered as last

Lungenforschung, DZL). The funders had no role in study design, data collection and analysis, decision to publish, or preparation of the manuscript.

**Competing interests:** The authors have declared that no competing interests exist.

**Abbreviations:** 6MWD, 6-minute walking distance; BMI, body mass index; COPD, chronic obstructive pulmonary disease; CF, cystic fibrosis; ILD, interstitial lung disease; LAS, lung allocation score; LTOT, long-term oxygen therapy; NIV, non-invasive ventilation; NPV, negative predictive value; NRT, nicotine replacement therapy; PPV, positive predictive value.

treatment option. However, despite advances in transplant medicine, long-term outcome is limited due to chronic allograft dysfunction and side effects of immunosuppression with an estimated median survival of 6.7 years after transplantation. [1]

Amongst others, both recipient and donor tobacco smoking have been shown to cause allograft dysfunction and mortality in solid organ transplant recipients. [2] Smoking resumption after lung transplantation increases the risk for allograft dysfunction, cancer and vascular disease resulting in reduced overall survival. [3–6] Therefore, smoking is regarded as contraindication for lung transplantation in most centers.

Smoking status is frequently assessed on the basis of self-reporting or clinical suspicion. [4] Research has shown that a high proportion of patients with lung disease falsely declare themselves to be non-smokers. [7] This can lead to an underestimation of smoking rates and to a reduced mutual trust of the treating physician. Similarly, prior to transplantation smoking is mostly assessed by questionnaire. Objective measures of tobacco smoking could be useful in improving clinical management and counseling of patients with difficulties to quit. Due to its high sensitivity and specificity, measurement of cotinine in serum or urine is widely used for diagnosis of nicotine consumption. [8]

Several studies have addressed smoking resumption after lung transplantation. Recently, research has shown that despite the severity of illness and the knowledge that quitting would have important long-term benefits, there continues to be a high proportion of patients who resume smoking after transplantation. [9–11] Patients with chronic obstructive pulmonary disease (COPD) and a short duration of smoking cessation prior to lung transplantation were at greatest risk of smoking resumption after lung transplantation. [4]

However efforts should be made to identify patients at risk before transplantation, since smoking cessation strategies can be applied and resumption potentially prevented. The aim of the study was to describe a systematic cotinine-based assessment of transplant candidates at different stages prior to lung transplantation.

## Methods

### Patients and study design

All lung transplant candidates from January 2017 until June 2019 were included in the study and analysed retrospectively (since January 2017 systematic cotinine-based screening was implemented at our transplant center). The study was conducted at the University of Munich, Germany and approved by the local ethics committee (UE No. 19–346; Ethics Commission of the Faculty of Medicine at the Ludwig-Maximilians-University Munich). Ethics committee waived the requirement for informed consent since data acquisition was retrospective and observational, data were anonymized and the study relied on measurements as part of routine care.

Upon referral, all lung transplant candidates are seen in our outpatient clinic to provide detailed information for the patient and to assess indications and potential contraindications for lung transplantation. After that, patients are discussed in the multidisciplinary conference. In the case of known absolute contraindications, patients are either rejected for transplantation or reevaluation at a later time point is recommended. For potential candidates, full evaluation to identify comorbidities and contraindications is recommended as reported previously. [12] Upon that, patients are discussed once more in the multidisciplinary team and placement on the waiting list or close monitoring in our center is recommended.

For the study patients were classified according to the stage prior to transplantation:
Stage 1: referral, i.e. first visit at the transplant center

Stage 2: time after first visit until placement on the waiting list, including patients during transplant evaluation and patients who are deemed too early and are closely monitored on a regular base in the center

Stage 3: patients actively listed for lung transplantation

Severity of disease was assessed by lung function analysis including blood gas analysis, spirometry, plethysmography and 6-minute walking distance (6MWD). The clinical course from referral to our center to the end of study period or lung transplantation was assessed retrospectively and data were obtained from medical records.

Underlying diseases were categorized into interstitial lung disease (idiopathic pulmonary fibrosis, non-specific interstitial pneumonia, cryptogenic pneumonia, hypersensitivity pneumonitis, fibrosis secondary to connective tissue disease), COPD (incl. alpha-1-antitrypsin deficiency), cystic fibrosis, and others. [13]

## Smoking status, cotinine-based screening, psychiatric comorbidity and addiction

Upon referral, i.e. first visit at our center, all patients were advised to avoid active and second hand smoking as well as nicotine replacement products. All patients were informed that to be nicotine-free for a minimum of 6 months before being placed on the waiting list is mandatory, since smoking resumption is associated with unfavorable outcome after transplantation. Furthermore, patients were informed that in the case of nicotine use, support can be provided and that regular cotinine tests are performed. In the case of false self-report, exclusion from our center may result.

Smoking status, including active tobacco and/or electric cigarette use, second hand smoking and the use of nicotine-replacement products was determined on the basis of self-reports assessed in interviews and was biochemically validated by a negative urinary or blood cotinine test at every visit in our center. Patients in evaluation/preparation for transplantation or patients on the waiting list are seen at least every 3 months in the transplant center. Urinary or serum cotinine was measured as a marker of active smoking or the use of nicotine-based products. Urinary and serum cotinine levels were assessed quantitatively by gas chromatography and mass spectroscopy (Agilent Technologies, GC/MS; GC-module G1530A, MS-module G1098A, Santa Clara, California, USA). Based on urinary or serum cotinine levels, patients were categorized positive in the case of a urinary level of >50ng/ml or a serum level of >10ng/ml. [14,15]

Transplant candidates who never smoked, or who have smoked less than 100 cigarettes in their lifetime were classified as never smoker.

Medical records were screened for coexisting psychiatric comorbidities, other addictions and psychiatric treatment/findings. All patients evaluated for lung transplantation were seen by a specialist for psychiatry and psychotherapy. In case of nicotine abuse or other addictive behavior non-pharmacological and pharmacological interventions are offered or recommended.

## Statistical analysis

Continuous variables are presented as the mean ± standard deviation, with categorical variables summarized by frequency and percentage. Depending on normal distributions t-test or Mann-Whitney-Wilcoxon test was used to compare continuous variables. Chi square test was used to compare categorical variables. Stepwise binary logistic regression was applied to identify the associations of certain variables on false self-reports. $p < 0.05$ was considered statistically significant. Data were statistically analysed by SPSS version 24.0 (IBM SPSS, Armonk, NY) statistical software.

## Results

### Study cohort

In total, 620 lung transplant candidates were included in the study and 1306 cotinine tests were performed during the observation period, accounting for 2.1 ± 1.7 (range 1–10) tests per patient. Underlying diseases of the study cohort were: COPD (n = 257, 41.5%), ILD (n = 228, 36.8%), CF (n = 87, 14.0%) and others (n = 48, 7.7%). The majority of the patients were male (57.3%; n = 355) with a mean age of 53.8 ± 11.9 years and a mean body mass index (BMI) of 23.4 ± 4.9 kg/m$^2$. Common comorbidities are depicted in the supplementary material (S1 Table).

In total, 92 patients (14.8%) patients had at least one positive cotinine test (Fig 1). Patients with a positive test were older (58.7 ± 6.2 vs. 52.9 ± 12.4 years, p<0.001) and had a lower Lung Allocation Score (LAS) (33.6 ± 2.5 vs. 39.5 ± 13.2, p = 0.010). Furthermore, patients with a positive test were more likely to have COPD (80.4% vs. 34.7%, p<0.0001) and have more often psychiatric comorbidities or a psychiatric disorder in the history (21.7% vs 11.3%, p = 0.011) compared to patients with only negative tests (Table 1). Finally, patients with a positive test reported more pack years (40.3 ± 19.0 vs 29.5 ± 19.4, p<0.0001) and time of smoking cessation was shorter (42.3 ± 50.1 vs 101.7 ± 104.1 weeks, p<0.0001) as shown in Table 1. The majority of non-COPD patients (n = 13, 72.2%) with a positive test had a cessation time of less than one year.

In addition, 193 patients (31.1%) reported never smoking. All were tested negative. Only 4 patients with negative cotinine test admitted smoking or use of nicotine replacement several days prior to screening (Fig 1).

To assess risk factors for a positive cotinine test multivariable regression analysis was performed while controlling for age, sex, underlying lung disease, coexisting psychiatric comorbidities or a history of psychiatric disorders, pack years and cessation time. COPD as underlying disease (odd ratio 2.102, CI 1.110–3.981; p = 0.023), the number of pack years (odd ratio 1.014, CI 1.000–1.028; p = 0.047) and a time of less than 1 year of smoking cessation (odd ratio 2.413, CI 1.410–4.128; p = 0.001) were independently associated with a positive cotinine test. Coexisting psychiatric comorbidities or a history of psychiatric disorders were not statistically significant associated with a positive test (odd ratio 1.291, CI 0.681–2.447; p = 0.434).

Of all positive tested patients, only 11 patients (12.0%) admitted active smoking and 3 patients (3.3%) reported the use of nicotine replacement therapy as shown in Fig 1. 78 patients

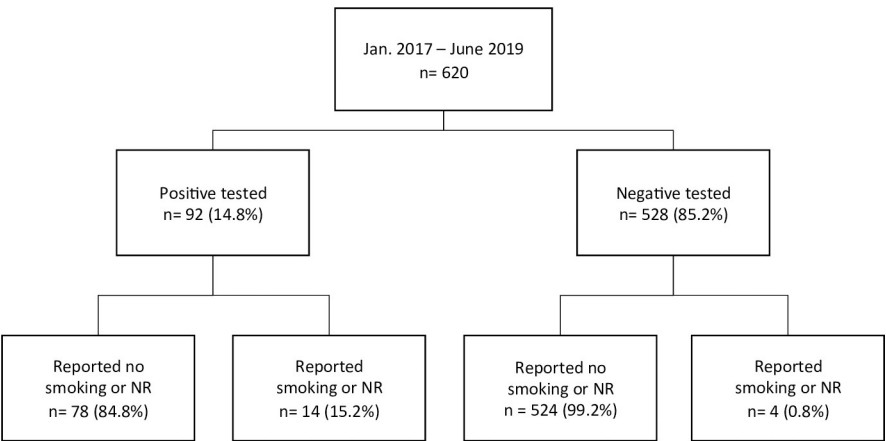

**Fig 1. Classification according to cotinine results and self-reports.** Data are presented as number and percentage, respectively. NR, nicotine replacement.

**Table 1. Characteristics of lung transplant candidates according to test results.**

|  | Patients with positive tests (n = 92) | Patients with negative tests (n = 528) | p-value |
|---|---|---|---|
| Age (years) | 58.7 ± 6.2 | 52.9 ± 12.4 | 0.000 |
| Sex (male), n (%) | 57 (62.0) | 298 (56.4) | 0.362 |
| BMI (kg/m$^2$) | 22.3 ± 4.7 | 23.6 ± 4.9 | 0.017 |
| Underlying diseases |  |  |  |
| COPD, n (%) | 74 (80.4) | 183 (34.7) | 0.000 |
| ILD, n (%) | 14 (15.2) | 225 (42.6) | 0.000 |
| CF, n (%) | 1 (1.1) | 86 (16.3) | 0.000 |
| Others, n (%) | 3 (3.3) | 34 (6.4) | 0.235 |
| Smoking history |  |  |  |
| Former smoker | 76 (82.6) | 337 (63.8) | 0.000 |
| Packyears, n | 40.3 ± 19.0 | 29.5 ± 19.4 | 0.000 |
| <1 year smoking cessation, n (%) | 64 (80.0) | 75 (23.1) | 0.000 |
| Time of smoking cessation (weeks) [a] | 42.3 ± 50.1 | 101.7 ± 104.1 | 0.000 |
| Lung Allocation Score | 33.6 ± 2.5 | 39.5 ± 13.2 | 0.010 |
| Psychiatric disorder, n (%) | 20 (21.7) | 60 (11.3) | 0.011 |

Data are presented as number and percentage, respectively.

[a] Available in 404 patients (80/324). BMI, body mass index; CF, cystic fibrosis; COPD, chronic obstructive pulmonary disease; ILD, interstitial lung diseases.

(84.7%) falsely declared not consuming any nicotine-based products prior to the test. Of those, the cause of positive cotinine could not be assessed.

Recurrent positive tests occurred in 12 patients (13.0%) with 8 patients (66.7%) suffering from COPD, 3 patients (25.0%) from ILD and 1 patient (8.3%) from pulmonary hypertension. 10 patients (83.3%) with a recurrent positive test declared falsely not using nicotine-based products.

The majority of patients with a positive test were patients at stage 1 (n = 53, 57.6%), i.e. at the first presentation in our center. However, 29 patients (31.5%) were already in stage 2 and 10 patients (10.9%) in stage 3, i.e. in preparation for listing and on the waiting list respectively. They were tested positive despite detailed information about smoking regulations at our center. The distribution of tests according to transplant stage is shown in Table 2. Positive tested patients are depicted in Fig 2 according to cotinine values and pre-transplant stage. Whereas medium serum cotinine decreased over the stages, no clear signal was found for urine cotinine

**Table 2. Distribution of tests and patients according to transplant stage.**

| Stage | All 1–3 | 1 | 2 | 3 |
|---|---|---|---|---|
| Number of test (%) | 1306 | 272 (20.8) | 551 (42.2) | 483 (37.0) |
| Number of patients (%) | 620 | 272 (43.7) | 361 (58.2) | 186 (30.0) |
| Positive tested patients, n (%[*]) | 92 (14.8) | 53 (19.5) | 29 (8.0) | 10 (5.4) |
| Self-reported smoking, n (%[#]) | 11 (12.0) | 8 (15.1) | 3 (10.3) | 0 (0.0) |
| Self-reported NR, n (%[#]) | 3 (3.3) | 1 (1.9) | 1 (3.4) | 1 (10.0) |
| False self-report, n (%[#]) | 78 (84.8) | 44 (83.0) | 25 (86.2) | 9 (90.0) |

Data are presented as number and percentage, respectively. NR, nicotine replacement.

[*] percent of all tested patients.

[#] percent of positive tested patients.

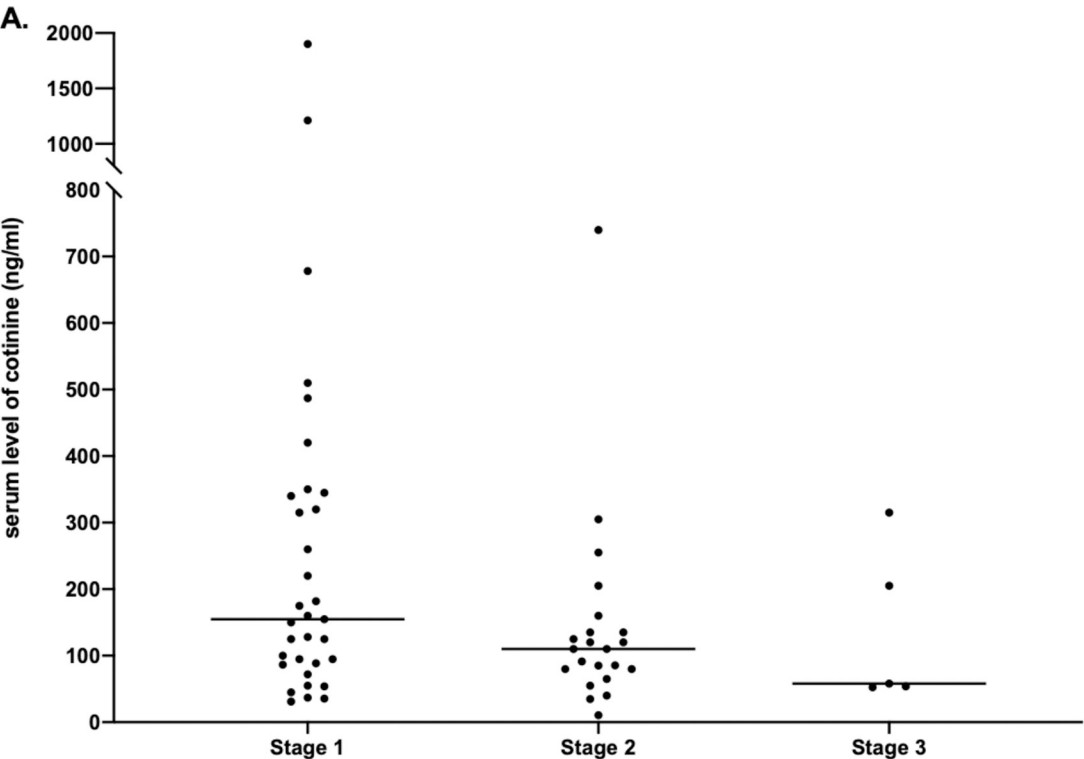

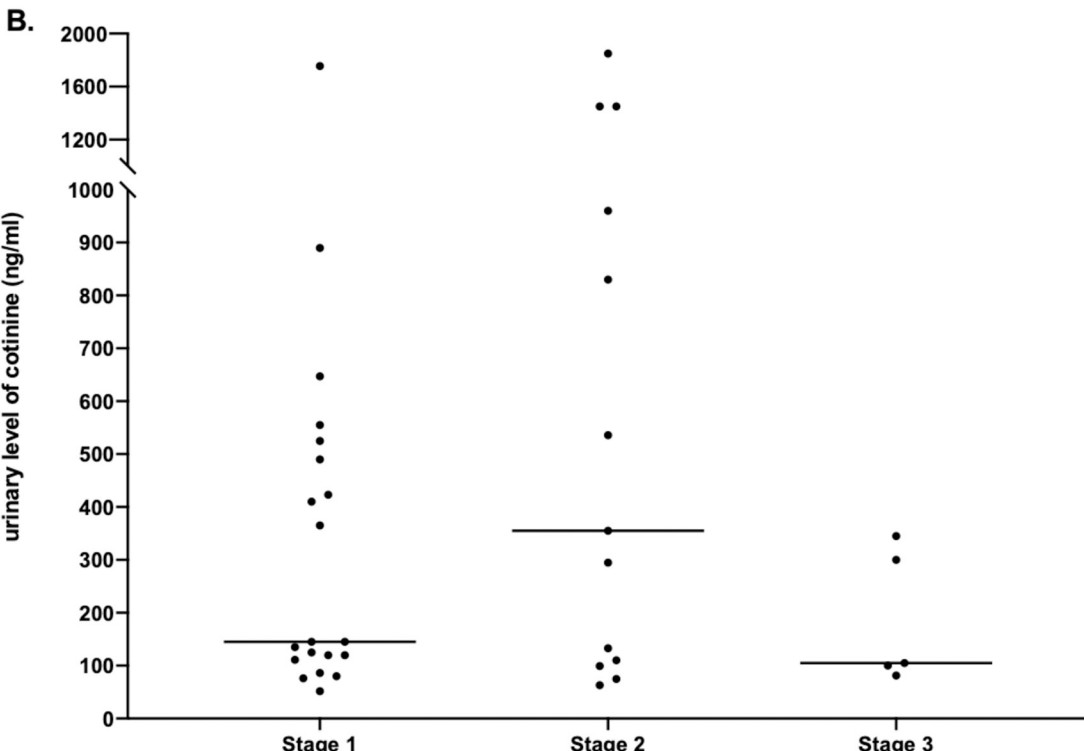

**Fig 2. Distribution of cotinine values over stages 1–3.** (A) serum levels of cotinine; (B) urinary levels of cotinine. The horizontal lines represent the medium values for each stage.

levels. However, the medium cotinine values were lowest in stage 3 in both, serum (58.0 ng/ml [53.4–260.0]) and urine (105.0 ng/ml [90.8–322.5]) as shown in Fig 2.

Furthermore, calculations for sensitivity, specificity, negative predictive value (NPV), and positive predictive value (PPV) over different stages prior to lung transplantation were performed (Fig 3). Specificity (true negative reports) was high and remained stable over the stages. Sensitivity (true positive reports) was low, however, decreased towards transplantation, suggesting that the number of cotinine positive patients falsely declaring nicotine abstinence increased over the stage, towards transplantation. If a patients reported active nicotine use the test was likely positive (PPV) and if a patient declared nicotine abstinence the test was likely negative (NPV).

## Discussion

Smoking in solid organ transplant recipients is associated with an increase in graft loss, cardiovascular events, malignancy, and mortality and is therefore regarded as absolute contraindication for lung transplantation. [16–19] Active smoking is a risk factor for early resumption after transplantation. [20] Therefore, attempts should be made to identify lung transplant candidates with active smoking behavior to provide support for smoking cessation or to not place patients on the waiting list. Self-reported smoking status may be particularly prone to false reports, since exclusion from transplantation may be life threatening in selected patients. [4]

In our cross-sectional observational study almost 15% of lung transplant candidates were tested positive for cotinine, indicating active smoking or nicotine replacement therapy. Most of the patients were positive at the first presentation at our center. However, patients during evaluation, follow-up, or even on the waiting list have been tested positive, despite detailed explanation about strict smoking regulations.

Approximately 85% of patients who smoke or use nicotine products do not accurately self-report use, supporting the previous finding that active smoking in underreported by patients with chronic lung disease. [7] In our study, the number of false self-reports of active nicotine use was

| All stages (1-3) | Patients with positive tests (n=92) | Patients with negative tests (n=528) |
|---|---|---|
| Smoking or NR reported (n=18) | 14 | 4 |
| No Smoking or NR (n=602) | 78 | 524 |

Sensitivity: 15.2%    Specificity: 99.2%    PPV:  77.7%    NPV: 87.0%

| Stage 1 | Patients with positive tests (n=53) | Patients with negative tests (n=219) |
|---|---|---|
| Smoking or NR reported (n=12) | 9 | 3 |
| No Smoking or NR (n=260) | 44 | 216 |

Sensitivity: 17.0%    Specificity: 98.6%    PPV: 75.0 %    NPV: 83.1%

| Stage 2 | Patients with positive tests (n=29) | Patients with negative tests (n=332) |
|---|---|---|
| Smoking or NR reported (n=5) | 4 | 1 |
| No Smoking or NR (n=356) | 25 | 331 |

Sensitivity: 13.8%    Specificity: 99.7%    PPV: 80%    NPV: 93.0%

| Stage 3 | Patients with positive tests (n=10) | Patients with negative tests (n=176) |
|---|---|---|
| Smoking or NR reported (n=1) | 1 | 0 |
| No Smoking or NR (n=185) | 9 | 176 |

Sensitivity: 10%    Specificity: 100%    PPV: 100%    NPV: 95.1%

**Fig 3. Accuracy of reporting nicotine use of transplant candidates.** Definition of abbreviation. PPV: positive predictive value; NPV negative predictive value.

particularly high, most likely due to the fact that the patients are aware of direct consequences, including postponing listing or ultimately exclusion from transplantation permanently. This is a potentially live threatening situation, which might force the patient to a false self-report. In this context, the number true positive reports decreased over the pre-transplant stages, suggesting that the number of cotinine positive patients falsely declaring nicotine abstinence increased towards transplantation. However, the numbers of positive tested patients is low, therefore a proper statistical comparison is not possible and interpretation should be performed cautiously.

COPD as underlying disease was identified as a risk factor for a positive cotinine test. However, our data demonstrated that also non-COPD patients can be affected and should be screened on a regular base. The number of individuals was too low to prove an association, but the majority of non-COPD patients had smoking cessation time less than one year, which should draw attention in affected patients.

In this line, the number of pack years and a duration of smoking cessation of less than one year were independently associated with a positive cotinine test in the entire cohort. This is in accordance with previous reports, demonstrating that a short duration of smoking cessation before transplantation is a risk for tobacco use after lung transplantation. [10]

On the other hand all never smoker were tested negative. Therefore, regular cotinine test in never smoker did not provide additional information and could be waived.

Psychiatric disorder or a history of psychiatric disorders, including addictive behavior were more often found in patients with a positive cotinine test compared to respective controls. However, in multivariable analysis an independent association could not be confirmed. Previous reports have shown that patients with psychiatric disorders demonstrate greater rates of tobacco use and nicotine dependence and quitting rates are lower. [21] Lung transplant candidates with a psychiatric and addictive history might benefit from targeted cessation intervention.

The results of our study should be interpreted in view of the study design and its limitations, which include a single-center cross-sectional study. Furthermore, despite providing data of a systematic cotinine screening we cannot prove that our approach would influence smoking resumption after transplantation. Furthermore, cotinine as biomarker does not allow to distinguish between surreptitious use of combustible tobacco versus pharmacologic use of NRT. In this context, exhaled carbon monoxide (CO) can be measured in expired air or in the blood, carboxyhemoglobin (COHb) in the blood using spectrophotometry. Both levels are highly correlated. [22,23] CO has limited sensitivity in detection of light smoking because CO levels from smoking are low and can be influenced by environmental sources (i.e. air pollution, open fires). [4,24] However, CO in addition to continine test could be used to distinguish between tobacco exposure and NRT. Several other tobacco exposure biomarkers, have been reported but may not always be practical to measure. [25]

In conclusion, our data demonstrate that lung transplant candidates who smoke or use nicotine products do not accurately self-report use. COPD was the most frequent diagnosis in affected patients. Patients with a history of heavy smoking and a short cessation time are at particular risk. Therefore, in patients at risk for active nicotine use a systematic cotinine based screening might improve optimal candidate selection and preparation for transplantation, which has to be proven in prospective controlled trials.

## Supporting information

**S1 File. Anonymized data set.**
(SAV)

**S1 Table. Comorbidities of the study cohort.**
(DOCX)

## Author Contributions

**Conceptualization:** Tobias Veit, Teresa Kauke, Jürgen Barton, Jürgen Behr, Nikolaus Kneidinger.

**Data curation:** Tobias Veit, Dieter Munker, Gabriela Leuschner, Carlo Mümmler, Alma Sisic, Teresa Kauke, Maximilian Huber, Jürgen Barton, Nikolaus Kneidinger.

**Formal analysis:** Tobias Veit, Dieter Munker, Gabriela Leuschner, Daniela Eser-Valerie, Maximilian Huber, Jürgen Barton, Nikolaus Kneidinger.

**Investigation:** Dieter Munker, Michael Irlbeck, Sebastian Michel, Daniela Eser-Valerie, Nikolaus Kneidinger.

**Methodology:** Christian Schneider, Sebastian Michel, Bruno Meiser, Jürgen Behr, Nikolaus Kneidinger.

**Project administration:** Gabriela Leuschner, Christian Schneider.

**Resources:** Alma Sisic, Bruno Meiser.

**Supervision:** Gabriela Leuschner, Christian Schneider, Sebastian Michel, Daniela Eser-Valerie, Katrin Milger, Bruno Meiser, Jürgen Behr, Nikolaus Kneidinger.

**Validation:** Alma Sisic.

**Visualization:** Tobias Veit, Dieter Munker, Carlo Mümmler, Michael Irlbeck, Katrin Milger, Nikolaus Kneidinger.

**Writing – original draft:** Tobias Veit, Carlo Mümmler, Michael Irlbeck, Katrin Milger, Jürgen Behr, Nikolaus Kneidinger.

**Writing – review & editing:** Tobias Veit, Dieter Munker, Gabriela Leuschner, Carlo Mümmler, Alma Sisic, Teresa Kauke, Christian Schneider, Michael Irlbeck, Sebastian Michel, Daniela Eser-Valerie, Maximilian Huber, Jürgen Barton, Katrin Milger, Bruno Meiser, Jürgen Behr, Nikolaus Kneidinger.

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
