## [Decision Letter · Decision Letter 0]

18 Mar 2020

PONE-D-20-02858

High prevalence of false reporting of smoking in lung transplant candidates

PLOS ONE

Dear Dr. Kneidinger,

Thank you for submitting your manuscript to PLOS ONE. After careful consideration, we feel that it has merit but does not fully meet PLOS ONE’s publication criteria as it currently stands. Therefore, we invite you to submit a revised version of the manuscript that addresses the points raised during the review process.

If you are interested in revising this paper, please pay close attention to the reviewer comments, both of which thought the paper topic had merit, but felt that the presentation and analyses presented were inadequate and required major revisions.   The topic of the paper is novel so we encourage the authors to consider how they might respond to reviewer suggestions for modifications to the paper.

We would appreciate receiving your revised manuscript by May 02 2020 11:59PM. To enhance the reproducibility of your results, we recommend that if applicable you deposit your laboratory protocols in protocols.io, where a protocol can be assigned its own identifier (DOI) such that it can be cited independently in the future. For instructions see: http://journals.plos.org/plosone/s/submission-guidelines#loc-laboratory-protocols

We look forward to receiving your revised manuscript.

Kind regards,

Michael Cummings, PhD

Academic Editor

PLOS ONE

Journal Requirements:

2. In ethics statement in the manuscript and in the online submission form, please provide additional information about the patient records used in your retrospective study. Specifically, please ensure that you have discussed whether all data were fully anonymized before you accessed them and/or whether the IRB or ethics committee waived the requirement for informed consent. If patients provided informed written consent to have data from their medical records used in research, please include this information.

"Disclosure: none".

a)    Please provide an amended Funding Statement that declares *all* the funding or sources of support received during this specific study (whether external or internal to your organization) as detailed online in our guide for authors at http://journals.plos.org/plosone/s/submit-now.  

b)    Please state what role the funders took in the study.  If any authors received a salary from any of your funders, please state which authors and which funder. If the funders had no role, please state: "The funders had no role in study design, data collection and analysis, decision to publish, or preparation of the manuscript."

"Competing Interests: none".

i) Please complete your Competing Interests on the online submission form to state any Competing Interests. If you have no competing interests, please state "The authors have declared that no competing interests exist.", as detailed online in our guide for authors at http://journals.plos.org/plosone/s/submit-now

ii)  This information should be included in your cover letter; we will change the online submission form on your behalf.

Reviewers' comments:

Reviewer's Responses to Questions

**Comments to the Author**

1. Is the manuscript technically sound, and do the data support the conclusions?

Reviewer #1: No

Reviewer #2: Partly

2. Has the statistical analysis been performed appropriately and rigorously? 

Reviewer #1: I Don't Know

Reviewer #2: No

3. Have the authors made all data underlying the findings in their manuscript fully available?

Reviewer #1: Yes

Reviewer #2: No

4. Is the manuscript presented in an intelligible fashion and written in standard English?

Reviewer #1: Yes

Reviewer #2: Yes

5. Review Comments to the Author

Reviewer #1: The authors used urinary cotinine levels to confirm tobacco abstinence among subjects being evaluated, and/or listed for lung transplantation. They report that active smoking (as evidenced by cotinine levels) is prevalent (~15%) in their lung transplant candidates, and that reliance upon self-reported abstinence is inaccurate for detecting tobacco use.

The study is well done, though I do not believe the study design allows them to faithfully backup their conclusions. Most importantly, though patients are restricted from using nicotine replacement therapy, it is possible (perhaps very likely) that many of the positive urinary screens could be related to individuals self-medicating with over-the-counter nicotine replacement therapy. What the finding of this paper illustrates to me is how profoundly addicting nicotine is, as evidenced by patient factors associated with positive nicotine test (those with psychiatric comorbidities, patients with heavier-pack-year exposure, and more recent quit dates). These factors are known to be associated with both higher prevalence of tobacco use, and greater risk of relapse. It is precisely these individuals, who are more susceptible to nicotine withdrawal symptoms, that might self-medicating with over-the-counter nicotine replacement products in order to avoid relapse to combustible tobacco.

Major:

These individuals should be supported in their attempts at tobacco cessation by any means necessary. Apparently their transplant program prohibits the use of nicotine replacement therapy, which is true of some other programs, but not universally the case for all transplant centers. Apart from the ethical issue of prohibiting use of an effective treatment for an addiction, this prohibition of nicotine replacement therapy probably limits the validity of their conclusions, in that the study as described could not reliably distinguish between surreptitious use of combustible tobacco versus pharmacologic use of NRT. There are alternative biomarkers which, though more expensive, are also more specific to combustible tobacco.

Minor:

the statement in the 2nd to final paragraph of the Results section indicates that the "...number of patients falsely declaring nicotine abstinence increased over the transplant stage" but the data shown in this regard in table 4 do not appear to have appropriate statistical confirmation. While the numbers do increase, the confidence intervals about these averages are likely too large to support a 'trend'.

Reviewer #2: Authors present an interesting manuscript on misreporting of smoking status in lung transplant patients. Data have some unique findings and the cohort represents the most recent analysis for this type of cohort. There are several areas where presentation of data in a different format would make information much more useful and clear. Standard calculations for sensitivity, specificity, NPV, and PPV would help as would a flowchart and breakdown of cotinine values. There are suggestions for reanalysis that could make this manuscript much more interesting and valuable, but the value of the data could be very high.

References for urinary and serum cotinine values should be noted.

Most of Table 1 is not useful to the manuscript. PFTs could be eliminated and simple description of age, sex, and BMI could be presented as a single line of text. Table 2 is also not particularly helpful and may be more useful as a supplementary table in lieu of suggestions below.

The presentation of results is somewhat confusing. The most interesting findings are in Table 4. It would be much more intuitive if the following were presented perhaps as a flowchart:

Number of patients screened

Number tested

Number with positive cotinine vs. negative cotinine

Number who reported no smoking who tested positive vs. number who reported smoking and tested positive AND number who reported no-smoking who tested negative vs. number who reported smoking who tested negative

How many patients reported never smoking according to cotinine results?

Authors should strongly consider a simple 2x2 table with calculations for sensitivity, specificity, NPV, and PPV according to self-report vs. cotinine positive. Consideration for these according to Stage 1-3 might be interesting as it appears these change closer to waitlisting.

In the results, the statement “patients with a positive test were more often former smokers (82.6% vs 63.8%, p<0.001)” does not make sense.

Throughout the manuscript, statements are slightly misleading regarding percentages. For example, in the 6th paragraph of the Results there is a statement “The number of patient falsely declaring nicotine abstinence increased over the transplant stage, with 83%, 86.2% and 90% in stage 1, 2 and 3, respectively.” I believe this should state “The number of cotinine positive patients falsely declaring nicotine abstinence…” These statements need correction throughout including in the abstract.

In Table 3, annotation “a” is not in the table.

The binary selection of positive cotinine values provides limited information. A scatterplot of cotinine values for serum and urine over Stage 1-3 would provide significant useful information to see distribution of values change over stage. This would provide an indirect representation of whether cotinine values are generally large, small, or somewhere in between.

The last paragraph of the Results is not particularly useful.

The final sentence of the abstract is not well supported. From the data, it appears that 85% of patients did not smoke with 81/92/95% of patients not smoking at Stage 1/2/3. Of patients who smoked, the accuracy did not appear to change much during stage. It is possible that patients who were never smokers (not clearly presented here) could be very accurate as would patients who report current smoking. Data suggest that quitting more than 1 year ago resulted in clearly more accuracy than quitting within the past year. Authors need to better consider the data for more representative conclusions.

It is anticipated that the above changes may result in more definitive discussion measures including where measuring cotinine is most useful.

6. PLOS authors have the option to publish the peer review history of their article (what does this mean?). If published, this will include your full peer review and any attached files.

Reviewer #1: No

Reviewer #2: No

---

## [Author Response · Author response to Decision Letter 0]

17 Apr 2020

Dear Editors, dear Reviewers, 

we thank both reviewers and the editor for reviewing our manuscript so carefully and raising important issues that helped us to improve the manuscript. To us, by presenting data in a different format and by clarifying misleading information, the manuscript is now more consistent and gives the reader a more thorough picture of our patients.

Sincerely, 

Nikolaus Kneidinger and co-authors

REVIEWER COMMENTS:

The full comments of the reviewers were the following:

Reviewer: #1

Comments to the Author

GENERAL

The authors used urinary cotinine levels to confirm tobacco abstinence among subjects being evaluated, and/or listed for lung transplantation. They report that active smoking (as evidenced by cotinine levels) is prevalent (~15%) in their lung transplant candidates, and that reliance upon self-reported abstinence is inaccurate for detecting tobacco use.

The study is well done, though I do not believe the study design allows them to faithfully backup their conclusions. Most importantly, though patients are restricted from using nicotine replacement therapy, it is possible (perhaps very likely) that many of the positive urinary screens could be related to individuals self-medicating with over-the-counter nicotine replacement therapy. What the finding of this paper illustrates to me is how profoundly addicting nicotine is, as evidenced by patient factors associated with positive nicotine test (those with psychiatric comorbidities, patients with heavier-pack-year exposure, and more recent quit dates). These factors are known to be associated with both higher prevalence of tobacco use, and greater risk of relapse. It is precisely these individuals, who are more susceptible to nicotine withdrawal symptoms, that might self-medicating with over-the-counter nicotine replacement products in order to avoid relapse to combustible tobacco.

MAJOR Comment:

These individuals should be supported in their attempts at tobacco cessation by any means necessary. Apparently their transplant program prohibits the use of nicotine replacement therapy, which is true of some other programs, but not universally the case for all transplant centers. Apart from the ethical issue of prohibiting use of an effective treatment for an addiction, this prohibition of nicotine replacement therapy probably limits the validity of their conclusions, in that the study as described could not reliably distinguish between surreptitious use of combustible tobacco versus pharmacologic use of NRT. There are alternative biomarkers which, though more expensive, are also more specific to combustible tobacco.

Response: We thank the reviewer for these important comments. 

Patients are informed that nicotine addiction is a risk factor for tobacco smoking resumption after transplantation and that nicotine replacement and active tobacco use are equally regarded is a contraindication. In the context of pre-transplant psychiatric evaluation patients are evaluated regarding addictive behaviour and in case non-pharmacological and pharmacological interventions are offered by our institution.

We agree that a clear differentiation between nicotine replacement and active tobacco use is not possible in our study. In total, 92 patients were tested positive and thereof 14 patients truly reported nicotine use (new Figure 1). Thereof, 11 patients reported tobacco and only 3 patients reported to use nicotine replacement products. We cannot conclude that the distribution is similar in patients falsely reporting not using nicotine-based products. However, this could indicate that among positive tested patients there is a considerable number of patients actively using tobacco. 

However, we completely agree that this is a limitation; therefor through out the manuscript the assumption of “false reporting smoking” was changed to “false reporting nicotine use” to avoid confusion. Furthermore, the limitation is discussed in the revised version of the manuscript.

In this line, as suggested, we agree with the referee that more sophisticated methods to directly detect active tobacco use in contrast to nicotine replacement products would facilitate patient selection and cessation in patients with addictive behavior. In particular the use of carbon monoxide (CO) or carboxyhemoglobin (COHb) is widely available and could be used to distinguish between active tobacco and nicotine replacement therapy. This is acknowledged in the revised version of the manuscript. 

Minor Comment:

the statement in the 2nd to final paragraph of the Results section indicates that the "...number of patients falsely declaring nicotine abstinence increased over the transplant stage" but the data shown in this regard in table 4 do not appear to have appropriate statistical confirmation. While the numbers do increase, the confidence intervals about these averages are likely too large to support a 'trend'.

Response: We thank both reviewers for this important comment. As suggested by referee #2 we performed 2x2 tables with calculations for sensitivity, specificity, negative predictive value (NPV), and positive predictive value (PPV) over different stages prior to lung transplantation (new Figure 3). 

Specificity (true negative reports) was high and remained stable over the stages. Sensitivity (true positive reports) was low, and decreased towards transplantation, suggesting that the number of cotinine positive patients falsely declaring nicotine abstinence increased over the stages, towards transplantation. If a patient reports active nicotine use the test is likely positive (PPV) and if a patient declares nicotine abstinence the test is likely negative (NPV). 

We agree with the reviewer that the number of patients, in particular of positive tested patients over the stages towards transplantation decrease, therefore, a proper statistical comparison is not possible and interpretation should be performed cautiously. This limitation is acknowledged in the revised version of the manuscript.

Reviewer: #2

Comments to the Author

Authors present an interesting manuscript on misreporting of smoking status in lung transplant patients. Data have some unique findings and the cohort represents the most recent analysis for this type of cohort. There are several areas where presentation of data in a different format would make information much more useful and clear. Standard calculations for sensitivity, specificity, NPV, and PPV would help as would a flowchart and breakdown of cotinine values. There are suggestions for reanalysis that could make this manuscript much more interesting and valuable, but the value of the data could be very high.

1/. References for urinary and serum cotinine values should be noted.

Response: We thank the reviewer for this request. As suggest, references for urinary and serum cotinine values were added to the revised version of the manuscript.

Jarvis, M.J., et al., Comparison of tests used to distinguish smokers from nonsmokers. Am J Public Health, 1987. 77(11): p. 1435-8. 

Benowitz, N.L., et al., Optimal serum cotinine levels for distinguishing cigarette smokers and nonsmokers within different racial/ethnic groups in the United States between 1999 and 2004. Am J Epidemiol, 2009. 169(2): p. 236-48.

2/. Most of Table 1 is not useful to the manuscript. PFTs could be eliminated and simple description of age, sex, and BMI could be presented as a single line of text. Table 2 is also not particularly helpful and may be more useful as a supplementary table in lieu of suggestions below.

Response: As suggest by the reviewer, the comments are addressed in the revised version of the manuscript. Table 1 was deleted from the manuscript and demographic characteristics are provided in the Results of the manuscript. Table 2 is provided as new Table S1 in the supplement material.

3/. The presentation of results is somewhat confusing. The most interesting findings are in Table 4. It would be much more intuitive if the following were presented perhaps as a flowchart:

Number of patients screened

Number tested

Number with positive cotinine vs. negative cotinine

Number who reported no smoking who tested positive vs. number who reported smoking and tested positive AND number who reported no-smoking who tested negative vs. number who reported smoking who tested negative

Response: We thank the reviewer for this comment. As suggested the results are presented as a flowchart. Since 2016 the German Medical Association request from all German lung transplant centers to provide evidence for nicotine abstinence in lung transplant candidates and suggest to routine use of cotinine measurements. Therefor, from all lung transplant candidates seen in our outpatient clinic from January 2017 until the end of the observation period (June 2019) cotinine tests were available.

New Figure 1

4/. How many patients reported never smoking according to cotinine results?

Response: In total, 193 patients (31.1%) reported never smoking, defined as a candidate who has never smoked, or who has smoked less than 100 cigarettes in his or her lifetime. All patients were tested negative. Therefore, routine testing in this never smokers seems not to be mandatory. Results are included and discussed in the revised version of the manuscript.

5/. Authors should strongly consider a simple 2x2 table with calculations for sensitivity, specificity, NPV, and PPV according to self-report vs. cotinine positive. Consideration for these according to Stage 1-3 might be interesting as it appears these change closer to waitlisting.

Response: We thank the reviewer for this important comment. We performed 2x2 tables with calculations for sensitivity, specificity, NPV, and PPV over different stages prior to lung transplantation (new Figure 3). Specificity (true negative reports) was high and remained stable over the stages. Sensitivity (true positive reports) was low, however, decreased towards transplantation, suggesting that the number of cotinine positive patients falsely declaring nicotine abstinence increased over the stage, towards transplantation. If a patient reports active nicotine use the test is likely positive (PPV) and if a patient declares nicotine abstinence the test is likely negative (NPV). 

The number of patients, in particular of positive tested patients over the stages towards transplantation decrease, therefore, a proper statistical comparison is not possible and interpretation should be performed cautiously.

6/. In the results, the statement “patients with a positive test were more often former smokers (82.6% vs 63.8%, p<0.001)” does not make sense.

Response: We thank the reviewer for this comment and apologize for confusion. We have deleted that part of the statement from the manuscript.

7/. Throughout the manuscript, statements are slightly misleading regarding percentages. For example, in the 6th paragraph of the Results there is a statement “The number of patient falsely declaring nicotine abstinence increased over the transplant stage, with 83%, 86.2% and 90% in stage 1, 2 and 3, respectively.” I believe this should state “The number of cotinine positive patients falsely declaring nicotine abstinence…” These statements need correction throughout including in the abstract.

Response: We thank both reviewers for this important comment and refer to request #5. Changes have been made as suggested throughout the revised manuscript.

8/. In Table 3, annotation “a” is not in the table.

Response: We apologize; the annotation “a” was added in the table (new Table 1). 

9/. The binary selection of positive cotinine values provides limited information. A scatterplot of cotinine values for serum and urine over Stage 1-3 would provide significant useful information to see distribution of values change over stage. This would provide an indirect representation of whether cotinine values are generally large, small, or somewhere in between.

Response: Thank you for this comment. As suggest the serum and urine cotinine values of positive tested patients are provided as a scatterplot over the stages in the revised version of the manuscript (Table 2 and new Figure 2). 

Whereas medium serum cotinine decreased over the stages, no clear signal was found for urine cotinine. However, the medium cotinine value was lowest in Stage 3 in both, serum and urine. This might indicate that active smoking or nicotine replacement is stopped before follow-up visit in the transplant center in patients active listed for lung transplantation. However, the number of tests, in particular of positive tests is low and decreased over the stages towards transplantation, therefore interpretation should be performed cautiously.

Results and limitations are incorporated and discussed in the revised version of the manuscript.

10/. The last paragraph of the Results is not particularly useful.

Response: Thank you for this comment. The paragraph was deleted from the manuscript.

11/. The final sentence of the abstract is not well supported. From the data, it appears that 85% of patients did not smoke with 81/92/95% of patients not smoking at Stage 1/2/3. Of patients who smoked, the accuracy did not appear to change much during stage. It is possible that patients who were never smokers (not clearly presented here) could be very accurate as would patients who report current smoking. Data suggest that quitting more than 1 year ago resulted in clearly more accuracy than quitting within the past year. Authors need to better consider the data for more representative conclusions.

Response: We agree with the referee that our statement is misleading and not supported by the presented data. As suggested the majority of the patients did report truly not to use nicotine based products and even some patients did report active tobacco use or nicotine replacement products. We agree that there are patients at risk, as assessed by regression analysis (cessation<1 year, high number of pack years, COPD) where regular cotinine tests might be useful. On the other hand, in never-smokers regular cotinine tests seem not to be mandatory. Therefore, our conclusion was revised accordingly and the sentence deleted. To avoid misinterpretation, we clarified that most patients report not to use nicotine based products and were consecutively tested negative. However, even the proportion of patients tested positive (14,8%) was low, the number seems to be high in the context of imminent lung transplantation. Furthermore, of patients tested positive there was a high prevalence of falsely declaring nicotine abstinence (84,8%). Accordingly, title and wording was revised throughout the manuscript. 

It is anticipated that the above changes may result in more definitive discussion measures including where measuring cotinine is most useful.

---

## [Decision Letter · Decision Letter 1]

19 May 2020

PONE-D-20-02858R1

High prevalence of falsely declaring nicotine abstinence in lung transplant candidates

PLOS ONE

Dear Dr. Kneidinger,

Thank you for submitting your revised manuscript to PLOS ONE. After careful consideration, we feel that it has merit.  However, before we can accept this paper in PLOS ONE we would like you to address the comments from the 2nd reviewer who requested some minor revisions to the discussion of the paper.   Therefore, we invite you to submit a revised version of the manuscript that addresses the points raised during the review process.

We would appreciate receiving your revised manuscript by Jul 03 2020 11:59PM. To enhance the reproducibility of your results, we recommend that if applicable you deposit your laboratory protocols in protocols.io, where a protocol can be assigned its own identifier (DOI) such that it can be cited independently in the future. For instructions see: http://journals.plos.org/plosone/s/submission-guidelines#loc-laboratory-protocols

We look forward to receiving your revised manuscript.

Kind regards,

Michael Cummings, PhD

Academic Editor

PLOS ONE

Reviewers' comments:

Reviewer's Responses to Questions

**Comments to the Author**

1. If the authors have adequately addressed your comments raised in a previous round of review and you feel that this manuscript is now acceptable for publication, you may indicate that here to bypass the “Comments to the Author” section, enter your conflict of interest statement in the “Confidential to Editor” section, and submit your "Accept" recommendation.

Reviewer #1: All comments have been addressed

Reviewer #2: (No Response)

2. Is the manuscript technically sound, and do the data support the conclusions?

Reviewer #1: Partly

Reviewer #2: Yes

3. Has the statistical analysis been performed appropriately and rigorously? 

Reviewer #1: Yes

Reviewer #2: Yes

4. Have the authors made all data underlying the findings in their manuscript fully available?

Reviewer #1: Yes

Reviewer #2: Yes

5. Is the manuscript presented in an intelligible fashion and written in standard English?

Reviewer #1: Yes

Reviewer #2: Yes

6. Review Comments to the Author

Reviewer #1: Though I am still concerned that the study design does not permit fully addressing the study question, I believe the authors have raised valid points and acknowledged the limitations of their data in support of the conclusions. IN doing so the paper has been strengthened and I support the authors interpretation of their findigs.

Reviewer #2: Results are much clearer and findings are very interesting. From the revised manuscript, the significant finding is that 85% of patients who smoked (or used NRT) did not report this. This is striking and I would offer that the opening to the Discussion could better reflect this finding. This is certainly more interesting than higher rates in COPD or pack years. I would recommend reorganizing the Discussion around this theme and opening with this finding, but otherwise much improved.

In the discussion, paragraph starting with “interestingly, approximately 85% of patients…” this sentence doesn’t make sense. I think the authors are trying to state “Approximately 85% of patients who smoke or use nicotine products do not accurately self-report use, supporting the previous finding that active smoking in underreported by patients with chronic lung disease”

In the final paragraph of the discussion, the first sentence appears incorrect and could perhaps conclude that while active smoking is present in a minority of patients, there is a high rate of misrepresentation that warrants biochemical confirmation, particularly in patients reporting an ever-smoking history or quitting smoking within the past year.

7. PLOS authors have the option to publish the peer review history of their article (what does this mean?). If published, this will include your full peer review and any attached files.

Reviewer #1: No

Reviewer #2: No

---

## [Author Response · Author response to Decision Letter 1]

21 May 2020

Dear Editors, dear Reviewers, 

we thank both reviewers and the editor for reviewing our manuscript once more. We have addressed the comments and adopted suggestions accordingly. To us, the manuscript now has improved further and gives the reader a thorough picture of our patients.

Sincerely, 

Nikolaus Kneidinger and co-authors

REVIEWER COMMENTS:

The full comments of the reviewers were the following:

Reviewer: #1

Though I am still concerned that the study design does not permit fully addressing the study question, I believe the authors have raised valid points and acknowledged the limitations of their data in support of the conclusions. IN doing so the paper has been strengthened and I support the authors interpretation of their findings.

Response: Thank you for your willingness to review and help to improve our manuscript. 

Reviewer #2: 

Results are much clearer and findings are very interesting. From the revised manuscript, the significant finding is that 85% of patients who smoked (or used NRT) did not report this. This is striking and I would offer that the opening to the Discussion could better reflect this finding. This is certainly more interesting than higher rates in COPD or pack years. I would recommend reorganizing the Discussion around this theme and opening with this finding, but otherwise much improved.

Response: We thank the referee for his comments to further improve our manuscript. As suggested we reorganized the Discussion accordingly. 

In the discussion, paragraph starting with “interestingly, approximately 85% of patients…” this sentence doesn’t make sense. I think the authors are trying to state “Approximately 85% of patients who smoke or use nicotine products do not accurately self-report use, supporting the previous finding that active smoking in underreported by patients with chronic lung disease”

Response: Thank you for this comment; we adopted the sentence as suggested. 

In the final paragraph of the discussion, the first sentence appears incorrect and could perhaps conclude that while active smoking is present in a minority of patients, there is a high rate of misrepresentation that warrants biochemical confirmation, particularly in patients reporting an ever-smoking history or quitting smoking within the past year.

Response: Thank for this suggestion. To avoid misinterpretation, the first sentence of the last paragraph was deleted and replaced by “In conclusion, our data demonstrate that lung transplant candidates who smoke or use nicotine products do not accurately self-report use….”

---

## [Editor Report · Decision Letter 2]

3 Jun 2020

High prevalence of falsely declaring nicotine abstinence in lung transplant candidates

PONE-D-20-02858R2

Dear Dr. Kneidinger,

We’re pleased to inform you that your manuscript has been judged scientifically suitable for publication and will be formally accepted for publication once it meets all outstanding technical requirements.

Kind regards,

Michael Cummings, PhD

Academic Editor

PLOS ONE
---

## [Editor Report · Acceptance letter]

9 Jun 2020

PONE-D-20-02858R2 

High prevalence of falsely declaring nicotine abstinence in lung transplant candidates 

Dear Dr. Kneidinger:

I'm pleased to inform you that your manuscript has been deemed suitable for publication in PLOS ONE. Congratulations! Your manuscript is now with our production department. 

Kind regards, 

on behalf of

Dr. Michael Cummings 

Academic Editor

PLOS ONE